# DECT Numbers in Upper Abdominal Organs for Differential Diagnosis: A Feasibility Study

**Fumihito Toshima \*** , **Norihide Yoneda, Kanako Terada, Dai Inoue and Toshifumi Gabata**

Department of Radiology, Kanazawa University Graduate School of Medical Sciences, 13-1, Takara-Machi, Kanazawa 920-8641, Ishikawa, Japan
\* Correspondence: fumihitotoshima@staff.kanazawa-u.ac.jp; Tel.: +81-76-265-2323

**Abstract:** Evaluating the similarity between two entities such as primary and suspected metastatic lesions using quantitative dual-energy computed tomography (DECT) numbers may be useful. However, the criteria for the similarity between two entities based on DECT numbers remain unclear. We therefore considered the possibility that a similarity in DECT numbers within the same organ could provide suitable standards. Thus, we assumed that the variation in DECT numbers within a single organ is sufficiently minimal to be considered clinically equivalent. Therefore, the purpose of this preliminary study is to investigate the differences in DECT numbers within upper abdominal organs. This retrospective study included 30 patients with data from hepatic protocol DECT scans. DECT numbers of the following parameters were collected: (a, b) 70 and 40 keV CT values, (c) slope, (d) effective Z, and (e, f) iodine and water concentration. The agreement of DECT numbers obtained from two regions of interest in the same organ (liver, spleen, and kidney) were assessed using Bland–Altman analysis. The diagnostic ability of each DECT parameter to distinguish between the same or different organs was also assessed using receiver operating characteristic analysis. The 95% limits of agreement within the same organ exhibited the narrowest value range on delayed phase (DP) CT [(c) −11.2–8.3%, (d) −2.0–1.5%, (e) −11.3–8.4%, and (f) −0.59–0.62%]. The diagnostic ability was notably high when using differences in DECT numbers on portal venous (PVP) and DP images (the area under the curve of DP: 0.987–0.999 in (c)–(f)). Using the variability in DECT numbers in the same organ as a criterion for defining similarity may be helpful in making a differential diagnosis by comparing the DECT numbers of two entities.

**Keywords:** dual-energy computed tomography; computed tomography attenuation curve; effective Z; iodine concentration; water concentration; DECT numbers; quantitative analysis

## 1. Introduction

Dual-energy computed tomography (DECT) is based on simultaneously obtaining two datasets at different energy levels [1,2]. Virtual monochromatic and metal decomposition images derived from DECT imaging techniques have proved useful for improving image quality (improving visibility) and reducing the radiation dose and contrast medium volume [2–8]. Thus, these images are becoming routinely used in clinical practice.

DECT allows tissue characterization using unique quantitative parameters derived from spectral imaging data. Thus, DECT is reportedly useful for various imaging diagnoses, such as differentiating acute intracranial hemorrhage from contrast staining [9], diagnosing acute pulmonary embolism [10], differentiating uric acid from non-uric acid calculi [11], diagnosing gout [12], diagnosing cholesteatoma [13], and diagnosing bone marrow edema [14–16]. Such DECT-based diagnostics are beginning to be applied clinically.

Furthermore, it has been reported that DECT numbers of quantitative parameters, including the slope of the spectral CT attenuation curve (slope), virtual non-contrast attenuation (VNC), effective atomic number (effective Z), iodine concentration (IC), and water concentration (WC), may be useful in differentiating malignant (especially metastases) from non-malignant lesions, such as metastatic lymph nodules from benign lymph nodules in patients

with breast cancer (cut-off value of slope, 5.1; sensitivity/specificity, 66/97.7%) [17], papillary thyroid carcinoma (cut-off value of slope, 5.1; sensitivity/specificity, 62.0/91.1%) [18], (cut-off value of normalized IC, 0.62; sensitivity/specificity, 62.5/85.7%) [19], gastric cancer (cut-off value of normalized IC, 0.333; sensitivity/specificity, 89.9/67.6%) [20], colorectal cancer (cut-off value of normalized effective Z, 0.881; sensitivity/specificity, 73.6/92.4%) [21], adrenal metastasis from adrenal adenoma (cut-off value of IC/VNC, 6.7; sensitivity/specificity, 95/95%) [22], and hepatic metastasis from hepatic abscess (cut-off value of normalized IC, 0.62; sensitivity/specificity, 78.6/88.9%) [23]. These literature-based values of DECT numbers for distinguishing metastases from non-metastatic lesions could be used in routine clinical practice. However, these numbers have not yet been applied to routine imaging workups, despite the need for a non-invasive diagnostic tool that can replace biopsy. One reason may be the issue of standardization; it may be difficult to apply literature-based values directly to clinical practice, as these quantitative DECT numbers may produce protean values, potentially influenced by several CT scanning factors (e.g., the CT vendor, scanner, scanning protocols, and contrast medium injection protocols). In addition, DECT numbers may vary from tumor to tumor, even for tumors in the same organs, and numbers of metastases may also depend on the characteristics of the primary tumors.

Whether or not a lesion is visually similar to the primary tumor remains of primary importance in distinguishing metastatic from non-metastatic lesions during routine imaging workups. Regarding quantitative DECT numbers, we also hypothesized that using the DECT number of the primary tumor (evaluating the similarity between the values of primary and suspected metastatic lesions) when the primary tumor is present would be preferable to using literature-based values (comparing the values of the target lesions with those of previous reports) in routine clinical practice. A recent study revealed that it was useful to evaluate the similarity between the primary tumor and suspected metastatic lymph node in patients with breast cancer [24]. This report may support our hypothesis. However, the criteria for the similarity between two entities based on DECT numbers remain unclear. We therefore considered the possibility that a similarity in DECT numbers within the same organ could provide a suitable standard. In other words, we assumed that the differences in DECT numbers within the same organ are small enough to be regarded as clinically similar. Thus, this preliminary study aimed to investigate the differences in DECT numbers within upper abdominal organs.

## 2. Methods

### 2.1. Study Population

The institutional review board at our hospital approved this retrospective preliminary study, and the requirement for informed consent from each patient was waived owing to the retrospective design. Using our institution's radiologic database between March 2019 and June 2020, 572 consecutive patients in whom DECT scanning with a multiphasic hepatic protocol was performed were identified. Patients were excluded for the following reasons: chronic hepatic diseases (chronic hepatitis or liver cirrhosis) and/or history of malignant hepatic tumors ($n = 539$); and insufficient concentration of contrast medium (<600 mg iodine/kg; $n = 3$). Thirty patients in whom DECT scanning with a multiphasic hepatic protocol was performed, and their upper abdominal organs (liver, spleen, and kidney) were likely to be healthy (Figure 1).

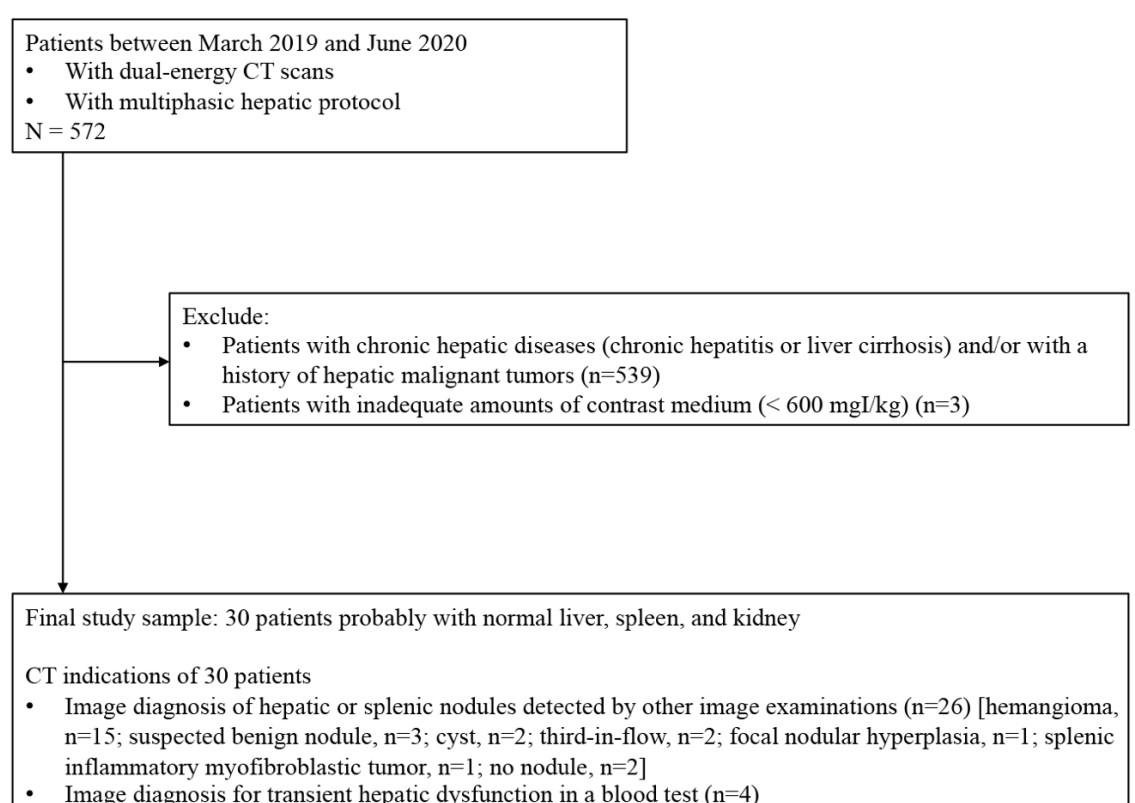

Patients between March 2019 and June 2020
- With dual-energy CT scans
- With multiphasic hepatic protocol
N = 572

Exclude:
- Patients with chronic hepatic diseases (chronic hepatitis or liver cirrhosis) and/or with a history of hepatic malignant tumors (n=539)
- Patients with inadequate amounts of contrast medium (< 600 mgI/kg) (n=3)

Final study sample: 30 patients probably with normal liver, spleen, and kidney

CT indications of 30 patients
- Image diagnosis of hepatic or splenic nodules detected by other image examinations (n=26) [hemangioma, n=15; suspected benign nodule, n=3; cyst, n=2; third-in-flow, n=2; focal nodular hyperplasia, n=1; splenic inflammatory myofibroblastic tumor, n=1; no nodule, n=2]
- Image diagnosis for transient hepatic dysfunction in a blood test (n=4)

**Figure 1.** Flow diagram of the study sample.

### 2.2. DECT Protocol

All fast kV-switching DECT examinations were performed using a 128 multi-detector row CT scanner (Revolution CT, GE Healthcare, Chicago, IL). The hepatic scanning protocol included non-contrast-enhanced and contrast-enhanced triple-phasic images. Contrast-enhanced images were obtained using a bolus-tracking technique. The hepatic arterial phase (HAP), portal venous phase (PVP), and delayed phase (DP) scans were performed 17, 51, and 137 s after thoracoabdominal aortic enhancement exceeded 200 HU, respectively. The CT scanning parameters were as follows: tube voltage, fast kV-switching (80 and 140 kV); noise index, 8.9 at 2.5-mm thickness (GSI Assist, GE Healthcare); iterative reconstruction, advanced statistical iterative reconstruction (ASiR-V, GE Healthcare) of 30%; slice thickness, 1.25 mm; slice interval, 1.25 mm, helical pitch, 0.508:1; contrast medium concentration, 300–370 mg iodine/mL; injection duration, 30 s; and injection dose, 600 mg iodine/kg.

### 2.3. Collection of DECT Numbers

Non-contrast-enhanced and triple-phasic contrast-enhanced CT images of 30 patients were transferred to the AW server (GE Healthcare). These multiphasic CT images were evaluated quantitatively by a radiologist with 12 years of experience in abdominal imaging using a GSI viewer (GE Healthcare). A GSI viewer automatically calculates various DECT numbers for the region of interest (ROI) when the ROI is placed on a 70-keV virtual monochromatic image. In this study, six quantitative DECT numbers composed of CT attenuation in (a) 70 and (b) 40 keV virtual monochromatic images (70 and 40 keV CT values), (c) slope, (d) effective Z, (e) IC, and (f) WC were collected for each non-contrast-enhanced and contrast-enhanced triple-phasic CT image. Circular ROIs with a diameter of 10 mm were placed on six locations, including the left hepatic lobe (ROI 1), right hepatic lobe (ROI 2), dorsal side of the spleen (ROI 3), ventral side of the spleen (ROI 4), right renal parenchyma (ROI 5), and left renal parenchyma (ROI 6), avoiding vessels, bile ducts, and the renal calyx (Figure 2). The slope was calculated using the following

formula: slope = (the difference between the mean CT attenuation at 40 and 70 keV virtual monochromatic images)/(the energy level difference [30 keV]). A total of 4,320 DECT numbers (30 patients × 4 phasic CT images × 6 DECT parameters × 6 locations) were collected. Among the data of 4320 DECT numbers, the data from bilateral renal parenchyma (ROI 5 and 6) on HAP CT images, in which the renal cortex and medulla were not clearly demarcated, were excluded, because it was impossible to place 10-mm ROIs on either only the cortex or the medulla. Therefore, the remaining 3960 DECT numbers were analyzed.

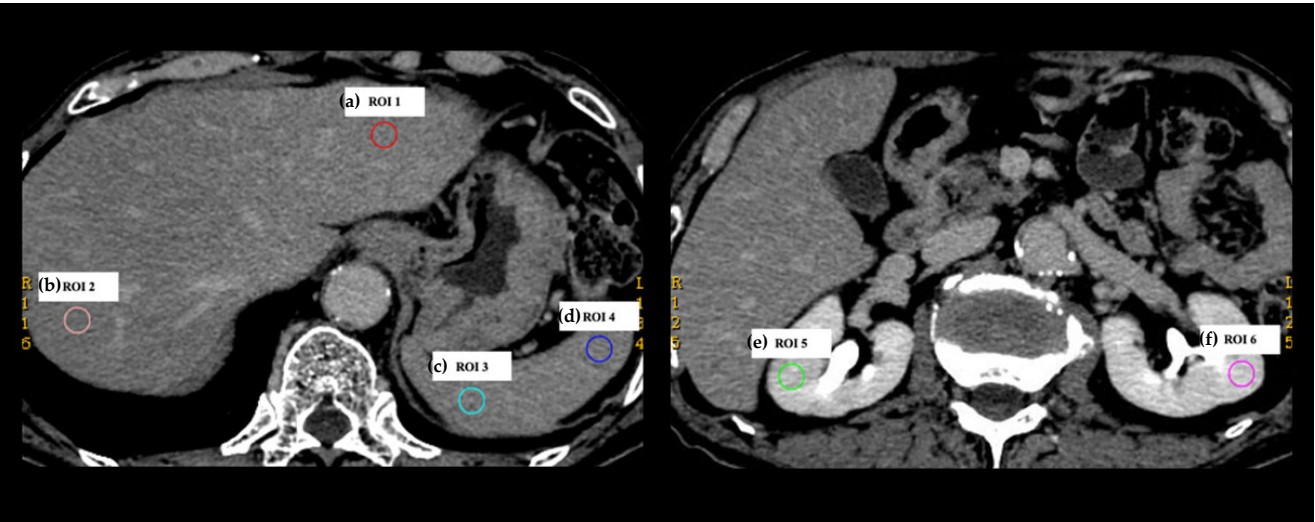

**Figure 2.** Placement of regions of interest (ROIs) on 70-keV virtual monochromatic image. Six ROIs were marked on each non-contrast-enhanced and contrast-enhanced triple-phasic CT image from 30 patients, respectively, avoiding vessels, bile ducts, and the renal calyx. The locations of the six ROIs were as follows: ROI 1, the left hepatic lobe; ROI 2, right hepatic lobe; ROI 3, the dorsal side of the spleen; ROI 4, the ventral side of the spleen; ROI 5, right renal parenchyma; and ROI 6, left renal parenchyma. Six DECT numbers, including CT attenuation in (**a**) 70 and (**b**) 40 keV virtual monochromatic images (70 and 40 keV CT values), (**c**) the slope of the spectral CT attenuation curve (slope), (**d**) effective atomic number (effective Z), (**e**) iodine concentration (IC), and (**f**) water concentration (WC), were automatically calculated for each ROI. As a result, 4,320 DECT numbers (30 patients × 4 phasic CT images × 6 DECT parameters × 6 locations) were collected.

### 2.4. Analyses of DECT Numbers

In this study, we examined two points: (i) the agreement (variability) of DECT numbers and ii) the diagnostic ability using the differences in DECT numbers. Regarding i), we assessed agreement in DECT numbers within the same abdominal organs (liver, spleen, or kidney) for each phasic CT. In other words, we evaluated the agreement between DECT numbers derived from ROI 1 and 2 (liver), ROI 3 and 4 (spleen), and ROI 5 and 6 (kidney), respectively. Regarding (ii), we assessed the diagnostic ability of each DECT parameter to differentiate measurements taken from the same organs and measurements taken from the different organs. Therefore, receiver operating characteristic (ROC) analysis was performed considering the differences in DECT numbers obtained from ROI 1 and 2 (liver), ROI 3 and 4 (spleen), and ROI 5 and 6 (kidney) as true (same organ); and the differences in DECT numbers obtained from ROI 1 and 3 (liver and spleen), ROI 3 and 5 (spleen and kidney), and ROI 5 and 1 (kidney and liver) as false (different organ).

### 2.5. Statistical Analyses

Statistical analyses were performed using GraphPad Prism 8 (GraphPad Software Inc., San Diego, CA, USA). Bland–Altman analysis was used to assess the agreement within the same healthy upper abdominal organs (between ROI 1 and 2, between ROI 3 and 4, and between ROI 5 and 6), using the difference (%) defined as (the difference in DECT numbers

derived from two ROIs)/(the average of DECT numbers derived from two ROIs) × 100. ROC analysis was performed to assess the diagnostic ability of each DECT parameter to differentiate the same organs from the different organs, by comparing the absolute difference in DECT numbers obtained from two ROIs within the same organs (ROI 1 and 2, ROI 3 and 4, and ROI 5 and 6) and the difference in DECT numbers obtained from two ROIs placed on different organs (ROI 1 and 3, ROI 3 and 5, and ROI 5 and 1). Additionally, pairwise comparisons of the area under the ROC curves (AUC) were performed using the Delong method. The absolute differences (%) were defined as [(the difference in DECT numbers obtained from two ROIs)/(the average of DECT numbers obtained from two ROIs)] × 100.

## 3. Results

### *3.1. Patient and CT Imaging Characteristics*

This study included the data of 9 men and 21 women, with a mean age of 59.4 ± 15.8 years (range, 22–80 years). Age was not associated with any DECT numbers. The indications for CT scanning were imaging diagnosis of hepatic or splenic nodules detected by other imaging examinations (e.g., ultrasonography and non-contrast-enhanced CT) in 26 patients, and screening for transient hepatic dysfunction in a blood test in the remaining four patients (Figure 1). No patients exhibited remarkable deformities, including enlargement or atrophy, of the liver, spleen, and kidney on CT images. Moreover, no patients exhibited severe fatty liver or hydronephrotic kidneys.

### *3.2. Agreement between Two DECT Numbers within the Same Organ*

Bland–Altman tests of the agreement between DECT numbers derived from two ROIs within the same organ (liver, spleen, and kidney) are shown in Figure 3 and Table 1. Among triple-phasic enhanced CT, the 95% limits of agreement (%) of all DECT parameters was the narrowest range for DP: (a) −9.5 to +7.8, (b) −9.7 to +7.3, (c) 11.2 to +8.3, (d) −2.0 to +1.5, (e) −11.3 to +8.4, and (f) −0.59 to +0.62, respectively.

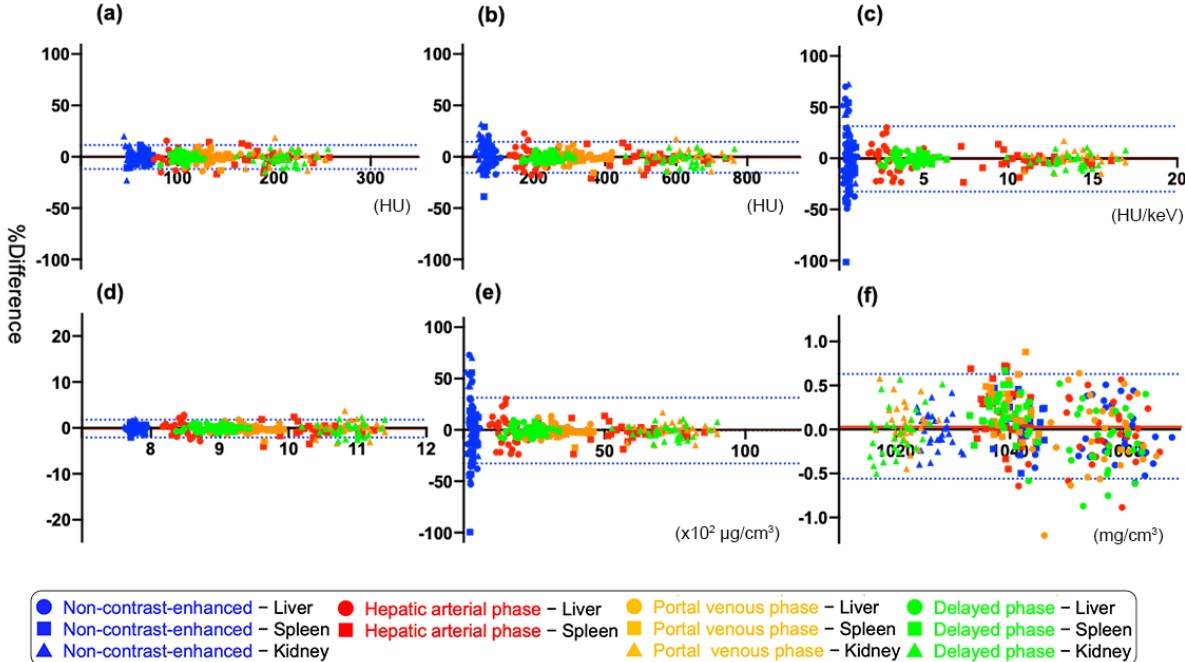

**Figure 3.** Bland-Altman tests showing the agreement between dual-energy CT (DECT) numbers obtained from two regions of interest within the same organ (liver, spleen, and kidney) in the six DECT parameters including (**a**) 70 keV CT value, (**b**) 40 keV CT value, (**c**) slope, (**d**) effective Z, (**e**) iodine concentration (IC), and (**f**) water concentration (WC). Each graph contains 330 plots, excluding the data of kidney on hepatic arterial images.

**Table 1.** Bland–Altman tests of the agreement between two ROIs within the same organ (liver, spleen, and kidney).

|  | Mean Bias (%) | 95% Limits of Agreement (%) |
|---|---|---|
| **(a) 70 keV CT value** (*n* = 330) * | 5.9 | −12.0, 11.3 |
| Non-contrast-enhanced (*n* = 90) | 7.3 | −14.0, 14.5 |
| HAP (*n* = 60) | 6.8 | −14.5, 12.0 |
| PVP (*n* = 90) | 5.1 | −9.9, 10.0 |
| DP (*n* = 90) | 4.4 | −9.5, 7.8 |
| **(b) 40 keV CT value** (*n* = 330) * | 7.7 | −15.7, 14.4 |
| Non-contrast-enhanced (*n* = 90) | 11.1 | −21.3, 22.0 |
| HAP (*n* = 60) | 8.6 | −18.4, 15.1 |
| PVP (*n* = 90) | 5.1 | −10.5, 9.6 |
| DP (*n* = 90) | 4.3 | −9.7, 7.3 |
| **(c) Slope** (*n* = 330) * | 16.3 | −32.6, 31.2 |
| Non-contrast-enhanced (*n* = 90) | 28.8 | −55.6, 57.5 |
| HAP (*n* = 60) | 11.5 | −24.3, 20.7 |
| PVP (*n* = 90) | 5.4 | −11.4, 9.9 |
| DP (*n* = 90) | 5.0 | −11.2, 8.3 |
| **(d) Effective Z** (*n* = 330) * | 1.0 | −2.1, 1.8 |
| Non-contrast-enhanced (*n* = 90) | 0.7 | −1.4, 1.4 |
| HAP (*n* = 60) | 1.4 | −3.0, 2.3 |
| PVP (*n* = 90) | 1.0 | −2.1, 1.8 |
| DP (*n* = 90) | 0.9 | −2.0, 1.5 |
| **(e) IC** (*n* = 330) * | 16.3 | −32.7, 31.3 |
| Non-contrast-enhanced (*n* = 90) | 29.0 | −55.9, 57.8 |
| HAP (*n* = 60) | 11.5 | −24.3, 20.7 |
| PVP (n = 90) | 5.4 | −11.5, 9.9 |
| DP (*n* = 90) | 5.0 | −11.3, 8.4 |
| **(f) WC** (*n* = 330) * | 0.31 | −0.56, 0.63 |
| Non-contrast-enhanced (*n* = 90) | 0.27 | −0.52, 0.52 |
| HAP (*n* = 60) | 0.34 | −0.61, 0.72 |
| PVP (*n* = 90) | 0.32 | −0.55, 0.70 |
| DP (*n* = 90) | 0.31 | −0.59, 0.62 |

* Data of all phasic CT images [non-contrast-enhanced and triple-phasic (HAP, PVP, and DP) contrast-enhanced CT images]. DECT, dual-energy computed tomography; ROI, region-of-interest; HAP, hepatic arterial phase; PVP, portal venous phase; DP, delayed phase; IC, iodine concentration; WC, water concentration

### 3.3. Diagnostic Ability

To evaluate the diagnostic ability of each DECT parameter to determine if the organs were the same or different, comparison tests between the difference in DECT numbers derived from two ROIs within the same organs and those from two ROIs placed on different organs were performed. The ROC curves for each phasic CT image are shown in Figure 4 and Table 2.

On non-contrast-enhanced CT, the AUCs of all but one of the DECT parameters [(b) 40 keV CT value, (c) slope, (d) effective Z, and (e) IC] were lower than those of (a) 70 keV. Even on HAP, the AUCs of DECT parameters [(b), (c), (d), and (e)] were not significantly different from that of (a) 70 keV CT value. On the other hand, the AUC of the (a) 70 keV CT value was significantly lower than those of (c) slope, (d) effective Z, (e) IC, and (f) WC [(a) versus (c), $p = 0.03$; versus (d), $p = 0.02$; versus (e), $p = 0.03$; versus (f), $p < 0.01$; respectively] on PVP. In particular, the (f) WC showed the highest AUC (0.996), followed by (d) effective Z (0.977) and (e) IC (0.975). Moreover, on DP, the AUC of the (a) 70 keV CT values was significantly lower than those of (c) slope, (d) effective Z, (e) IC, and (f) WC [(a) versus (c), $p < 0.01$; versus (d), $p < 0.01$; versus (e), $p < 0.01$; versus (f), $p < 0.01$; respectively]. Moreover, the AUC of the (b) the 40 keV CT value was also significantly lower than those of (c) slope, (d) effective Z, (e) IC, and (f) WC [(b) versus (c), $p < 0.01$; versus (d), $p < 0.01$; versus (e), $p < 0.01$; versus (f), $p < 0.01$; respectively]. WC had the highest AUC (0.999), followed by (d) effective Z (0.988), (c) slope (0.987), and (e) IC (0.987).

**Table 2.** ROC analyses between the difference in DECT numbers obtained from two ROIs placed on the same organs and the difference in DECT numbers obtained from two ROIs placed on different organs.

| | | AUC | Cut-Off 1 * (%) | Sens (%) | Spec (%) | Cut-Off 2 * (%) | Sens (%) | Spec (%) |
|---|---|---|---|---|---|---|---|---|
| Non-contrast-enhanced | (a) 70 keV CT value | 0.980 (0.963, 0.996) | 12.5 | 94.4 | 94.4 | 13.9 | 95.6 | 91.1 |
| | (b) 40 keV CT value | 0.837 (0.790, 0.884) | 19.1 | 93.3 | 67.8 | 21 | 95.6 | 66.7 |
| | (c) Slope | 0.692 (0.626, 0.759) | 24.2 | 68.9 | 65.6 | 55.1 | 95.6 | 28.9 |
| | (d) Effective Z | 0.717 (0.656, 0.779) | 0.65 | 72.2 | 65.6 | 1.34 | 95.6 | 32.2 |
| | (e) IC | 0.689 (0.623, 0.756) | 23.4 | 68.9 | 65.6 | 56.4 | 95.6 | 26.7 |
| | (f) WC | 1.000 (0.999, 1.000) | 0.66 | 100 | 98.9 | 0.51 | 95.6 | 100 |
| HAP | (a) 70 keV CT value | 0.953 (0.928, 0.977) | 16.8 | 100 | 81.1 | 15.4 | 95 | 83.3 |
| | (b) 40 keV CT value | 0.958 (0.936, 0.980) | 20.9 | 98.3 | 87.8 | 18.2 | 95 | 88.9 |
| | (c) Slope | 0.943 (0.915, 0.970) | 25.1 | 98.3 | 86.7 | 23.6 | 95 | 87.8 |
| | (d) Effective Z | 0.962 (0.940, 0.984) | 4.05 | 100 | 88.9 | 2.85 | 95 | 91.1 |
| | (e) IC | 0.942 (0.914, 0.970) | 25.4 | 98.3 | 86.7 | 23.7 | 95 | 87.8 |
| | (f) WC | 0.994 (0.988, 1.000) | 0.72 | 98.3 | 96.7 | 0.71 | 95 | 96.7 |
| PVP | (a) 70 keV CT value | 0.932 (0.903, 0.960) | 11.6 | 95.6 | 78.9 | 11.6 | 95.6 | 78.9 |
| | (b) 40 keV CT value | 0.963 (0.941, 0.985) | 14.2 | 97.8 | 90 | 12.8 | 95.6 | 90 |
| | (c) Slope | 0.975 (0.958, 0.992) | 17.2 | 100 | 92.2 | 13.9 | 95.6 | 93.3 |
| | (d) Effective Z | 0.977 (0.961, 0.992) | 3.79 | 100 | 90 | 2.71 | 95.6 | 92.2 |
| | (e) IC | 0.975 (0.959, 0.992) | 17.2 | 100 | 92.2 | 13.9 | 95.6 | 93.3 |
| | (f) WC | 0.996 (0.992, 1.000) | 0.88 | 98.9 | 96.7 | 0.64 | 95.6 | 97.8 |
| DP | (a) 70 keV CT value | 0.920 (0.887, 0.952) | 11.1 | 98.9 | 71.1 | 8.6 | 95.6 | 72.2 |
| | (b) 40 keV CT value | 0.947 (0.919, 0.974) | 13.1 | 98.9 | 87.8 | 10.1 | 95.6 | 90 |
| | (c) Slope | 0.987 (0.977, 0.997) | 12 | 98.9 | 93.3 | 10.5 | 95.6 | 95.6 |
| | (d) Effective Z | 0.988 (0.979, 0.996) | 2.55 | 98.9 | 92.2 | 2.22 | 95.6 | 92.2 |
| | (e) IC | 0.987 (0.977, 0.997) | 11.9 | 98.9 | 93.3 | 10.5 | 95.6 | 95.6 |
| | (f) WC | 0.999 (0.998, 1.000) | 0.72 | 98.9 | 97.8 | 0.61 | 95.6 | 100 |

* Cut-off value 1 was defined as the value when the sensitivity and specificity were the highest (Youden index), and a cut-off value 2 was the value when the sensitivity was set to 95%. DECT, dual-energy computed tomography; ROI, region-of-interest; HAP, hepatic arterial phase; PVP, portal venous phase; DP, delayed phase; IC, iodine concentration; WC, water concentration; AUC, area under the ROC curve; Sens, sensitivity; Spec, specificity.

The cut-off values at 95% sensitivity and at the highest sensitivity and specificity (Youden index) are also shown in Table 2.

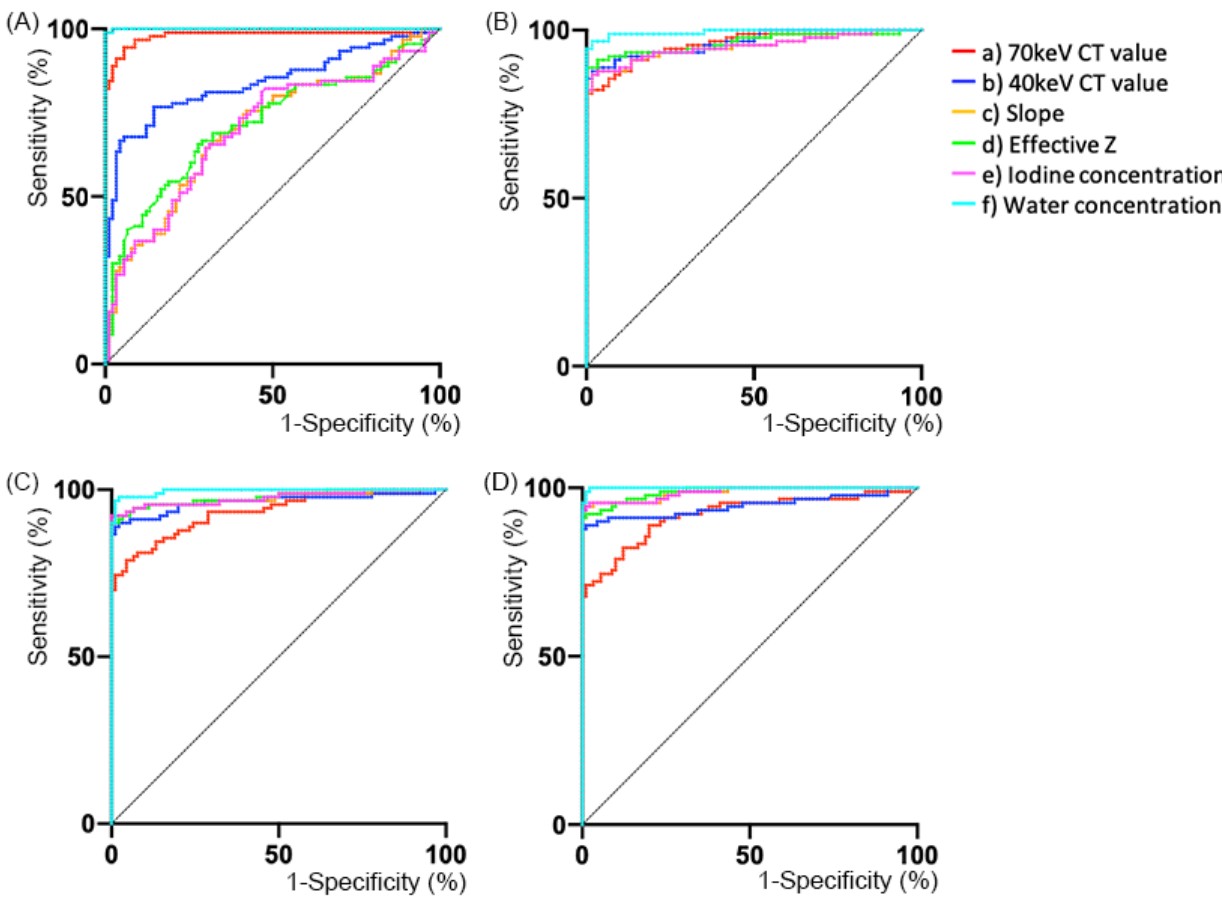

**Figure 4.** Receiver operating characteristic (ROC) analyses in non-contrast-enhanced (**A**) and contrast-enhanced CT including (**B**) hepatic arterial phase (HAP), (**C**) portal venous phase (PVP), and (**D**) delayed phase (DP) showing the comparisons between the difference in dual-energy CT (DECT) numbers derived from two regions of interest (ROIs) within the same organs (ROI 2–ROI 1 [liver], ROI 4–ROI 3 [spleen], and ROI 6–ROI 5 [kidney]), and the difference in DECT numbers derived from 2 ROIs placed on different organs (ROI 3–ROI 1 [spleen–liver], ROI 5–ROI 3 [kidney–spleen], and ROI 5–ROI 1 [kidney–liver]).

## 4. Discussion

DECT numbers of quantitative parameters have already been reported to be useful in differentiating malignant (especially metastases) from non-malignant lesions. For instance, it has been reported that a cut-off value of 3.2–3.67 for slope $_{\text{arterial phase}}$, 0.1–0.21 for normalized IC $_{\text{arterial phase}}$, 0.7–0.78 for normalized effective Z $_{\text{arterial phase}}$, 3.5–5.1 for slope $_{\text{portal or venous phase}}$, 0.33–0.62 for normalized IC $_{\text{portal or venous phase}}$, and 0.87–0.9 of normalized effective Z $_{\text{portal or venous phase}}$ result in good sensitivity and specificity in the evaluation of lymph node metastases [17–23]. These literature-based values of DECT numbers could be used clinically, however, they have not yet been applied to routine imaging workups, as these quantitative DECT numbers may be affected by several CT scanning factors and may vary from tumor to tumor, even for tumors in the same organs. We predicted that evaluating the similarity between DECT numbers of two entities would be more feasible than evaluating the difference between the numbers of the target lesion and literature-based values. The need to create a criterion for defining similarity led us to consider the difference (variability) in DECT numbers within the same organ. If the difference between the DECT numbers of two entities is only as large as the difference between the same organ, the two entities may be considered similar; thus, they are related (e.g., the target lesion is metastasis of the primary tumor).

This study investigated the variability in DECT numbers within the upper abdominal organs (liver, spleen, and kidney). Among multi-phasic contrast-enhanced CT, the variability of DECT numbers on DP CT was the narrowest range: the 95% limits of agreement (%), approximately ±8.7 in (a) 70 keV CT value, ±8.5 in (b) 40 keV CT value, ±9.8 in (c) slope, ±1.8 in (d) effective Z, ±9.9 in (e) IC, and ±0.61 in (f) WC, respectively. This study also revealed the diagnostic ability to distinguish between the same or different organs in order to predict which DECT parameters would be useful clinically in each phasic of CT images. On non-contrast-enhanced and HAP contrast-enhanced CT, no parameter showed higher diagnostic ability than 70 keV CT, which is generally treated as equivalent to conventional 120 kV single-energy CT [25], except for WC. Conversely, 40 keV CT value, slope, effective Z, IC, and WC had higher diagnostic capacity than 70 keV CT value on PVP and DP images. Given this result, it may be worthwhile evaluating the similarity using DECT numbers, especially on PVP or DP images for differential diagnosis in routine workups.

To our knowledge, few studies have investigated the similarity between two entities such as primary and suspected metastatic lesions, with the only one reported by Terada et al. [24]. This study clarified that the similarity between the DECT numbers of the primary tumor and suspected metastatic lymph node was more useful than the simple DECT numbers. Furthermore, the best sensitivity and specificity (66.7 and 81.6%) for identifying lymph node metastases were obtained when the error in 40 keV CT values from the primary tumor was within 10.5%, and the best sensitivity and specificity (75.0 and 75.6%) for diagnosing small (<5 mm) lymph node metastases were achieved when the error was within 13.3%, which was compatible with the results of this study (the assessment of the variability in DECT numbers within the same organ and diagnostic ability to distinguish between the same or different organs). This recent report supports the hypothesis of our study. In other words, this report would encourage the clinical use of our preliminary results (e.g., difference in 40 keV CT values on DP within 13.1%, as described in Table 2) as a criterion for defining similarity for differential diagnosis.

This study had several limitations. First, there might have been a selection bias due to the retrospective design, despite the enrolled patients meeting the inclusion criteria. The observational design, inclusion of a single center, a single specific CT scanner, and relatively small sample size might also have affected the variability of the quantitative parameters. In addition, only the difference between two points was evaluated by placing small ROIs to avoid non-normal parenchyma such as lesions, vessels, bile ducts, and the renal calyx, and inter- or intra-observer agreements were not examined. We assumed that the differences within the same organ would be sufficiently small to be clinically irrelevant, although this was not pathologically confirmed. Finally, we did not evaluate performance on a test dataset. Additional multi-center studies with a larger number of patients and observers are needed.

## 5. Conclusions

Using the variability in DECT numbers in the same organ as a criterion for defining similarity may be helpful in making a differential diagnosis by comparing the DECT numbers of two entities.

## 6. Take Away in a Sentence

- The value ranges of DECT numbers within the same abdominal organs were particularly narrow on DP images.
- Diagnostic ability to distinguish between the same or different organs was notably high when using the differences in DECT numbers on PVP or DP images.

**Author Contributions:** Conceptualization, N.Y.; methodology, F.T. and N.Y.; formal analysis, F.T.; investigation, F.T. and K.T.; data curation, F.T.; writing—original draft preparation, F.T.; writing—review and editing, F.T.; visualization, F.T.; supervision, D.I. and T.G.; project administration, D.I.; funding acquisition, D.I. All authors have read and agreed to the published version of the manuscript.

**Funding:** This research received no external funding.

**Institutional Review Board Statement:** The study was conducted in accordance with the Declaration of Helsinki, and approved by the Institutional Review Board of Kanazawa University Hospital (protocol code, 113780-1; date of approval, 18 August 2021).

**Informed Consent Statement:** Patient consent was waived due to the retrospective design.

**Data Availability Statement:** Not applicable.

**Acknowledgments:** We thank Kosuke Sasaki of GE Healthcare for the support of data collection.

**Conflicts of Interest:** The authors declare no conflict of interest.

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
