# Peer review of "DECT Numbers in Upper Abdominal Organs for Differential Diagnosis: A Feasibility Study"

_tomography, doi:10.3390/tomography8060225_

Round 1

Reviewer 1 Report

This is the review of the paper

Can dual-energy computed tomography data-based quantitative parameters be used for differential image diagnosis? A 3 preliminary study on the variability of dual-energy computed  tomography parameters in healthy upper abdominal organs

TITLE

Is too long; I would suggest to focus on main topic (for example:

DECT numbers in upper abdominal organs for differential diagnosis: a feasibility study)

ABSTRACT

Purpose: the background is unclear: I suggest the possibility of English editing in this section to focus on main topic and give rationale of study.

Also, regarding background I suggest to include the use in several previous studies of DECT numbers for differential diagnosis (for example for detecting bone marrow edema in skeletal system, or to identify cholesteatoma in chronic otitis). Also DECT numbers were proposed for the assessment of treatment response.

Methods and results are overall described here.

Purpose and conclusion should match those in intro and discussion section

KEY words

I would add DECT numbers and quantitative analysis

INTRODUCTION

Background is not sufficiently described.

I would give a larger pictures of DECT clinical approved uses, especially in MSK imaging, including papers dealing with the diagnosis of BME, and the evaluation of prosthesis.

The first topic, also, is important because there are some papers using DECT numbers with relative cut-off for differential diagnosis. These papers could be briefly described in the introduction, included in reference list, and eventually discussed in the discussion section.

After improving the background, I suggest to rephrase the second part of intro section, focusing on rationale of paper.

For example

Some literature-based values of DECT numbers could be used in routine clinical practice for distinguish tumor and metastases from normal tissues and benign lesions.

Rationale is relatively strong to me and should be clarified.

Purpose should be rephrased to be more direct and concise and should be reported at the end of abstract.

METHODS

Ok

RESULTS

Are overall well presented

To help the reader give main results in the text and leave details in tables.

DISCUSSION

The authors described again results In details here. However my suggestion is to give main results and try a comparison and discussion with previous similar studies.

Also clinical implications od f results should be discussed here.

Among limitations I would include the use of a single specific scanner (DECT have several type of scanners from different vendors)

Conclusions are too long; please focus on my results

References

The list is relatively short; authors could search the literature and identify DECT study to give a larger pictures.

The papers to be considered, not in specific order, are the following.

D'Angelo T, Albrecht MH, Caudo D, Mazziotti S, Vogl TJ, Wichmann JL, Martin S, Yel I, Ascenti G, Koch V, Cicero G, Blandino A, Booz C. Virtual non-calcium dual-energy CT: clinical applications. Eur Radiol Exp. 2021 Sep 3;5(1):38. doi: 10.1186/s41747-021-00228-y. PMID: 34476640; PMCID: PMC8413416.   Yang Z, Zhang X, Fang M, Li G, Duan X, Mao J, Shen J. Preoperative Diagnosis of Regional Lymph Node Metastasis of Colorectal Cancer With Quantitative Parameters From Dual-Energy CT. AJR Am J Roentgenol. 2019 Jul;213(1):W17-W25. doi: 10.2214/AJR.18.20843. Epub 2019 Apr 17. PMID: 30995087.   Diagnostic accuracy of dual-energy CT and virtual non-calcium techniques to evaluate bone marrow edema in vertebral compression fractures G Foti, A Beltramello, M Catania, S Rigotti, G Serra, G Carbognin La radiologia medica 124 (6), 487-494     Identification of bone marrow edema of the ankle: diagnostic accuracy of dual-energy CT in comparison with MRI G Foti, M Catania, S Caia, L Romano, A Beltramello, C Zorzi, G Carbognin La radiologia medica 124 (10), 1028-1036   · Preoperative Prediction of Cervical Nodal Metastasis in Papillary Thyroid Carcinoma: Value of Quantitative Dual-Energy CT Parameters and Qualitative Morphologic Features. Wu YY, Wei C, Wang CB, Li NY, Zhang P, Dong JN.AJR Am J Roentgenol. 2021 May;216(5):1335-1343. doi: 10.2214/AJR.20.23516. Epub 2021 Mar 24.PMID: 33760651   · Clinical significance of dual-energy CT-derived iodine quantification in the diagnosis of metastatic LN in colorectal cancer. Kato T, Uehara K, Ishigaki S, Nihashi T, Arimoto A, Nakamura H, Kamiya T, Oshiro T, Ebata T, Nagino M.Eur J Surg Oncol. 2015 Nov;41(11):1464-70. doi: 10.1016/j.ejso.2015.08.154. Epub 2015 Aug 22.PMID: 26329783

Tables ok

Figures ok

Author Response

REVIEWER COMMENTS:

Reviewer #1:

This is the review of the paper 

Can dual-energy computed tomography data-based quantitative parameters be used for differential image diagnosis? A 3 preliminary study on the variability of dual-energy computed  tomography parameters in healthy upper abdominal organs

Manuscript ID: tomography-1957996

We hereby resubmit our manuscript titled, “DECT numbers in upper abdominal organs for differential diagnosis: a feasibility study.”

We sincerely appreciate all the comments and suggestions from the three reviewers as they have helped make our manuscript more precise and informative. We have read all the comments carefully and accordingly revised our manuscript.

Our point-by-point responses to each comment are noted below.

We really appreciate your kind consideration on this manuscript for publication in Tomography.

Comment 1

TITLE

Is too long; I would suggest to focus on main topic (for example: 

DECT numbers in upper abdominal organs for differential diagnosis: a feasibility study)

Response: We thank the reviewer for this suggestion. We have changed the title as follows: “DECT numbers in upper abdominal organs for differential diagnosis: a feasibility study.”

Comment 2

ABSTRACT

Purpose: the background is unclear: I suggest the possibility of English editing in this section to focus on main topic and give rationale of study.

Also, regarding background I suggest to include the use in several previous studies of DECT numbers for differential diagnosis (for example for detecting bone marrow edema in skeletal system, or to identify cholesteatoma in chronic otitis). Also DECT numbers were proposed for the assessment of treatment response.

Response: We thank the reviewer for this advice. We have re-written the background more clearly in the abstract section (lines 9-15). In addition, we have cited several previous studies on the differential diagnosis of various diseases using DECT in the introduction section, including bone marrow edema and cholesteatoma (lines 38-42).

Comment 3

Methods and results are overall described here.

Response: We thank the reviewer for this advice. The entire methods and results sections have been re-described in the abstract section (lines 15-25).

Comment 4

Purpose and conclusion should match those in intro and discussion section

Response: We thank the reviewer for this suggestion. In response to this, we have modified the manuscript to ensure consistency of content throughout.

Comment 5

KEY words

I would add DECT numbers and quantitative analysis

Response: We thank the reviewer for this comment. We have added the relevant description accordingly (line 29).

Comment 6

INTRODUCTION

Background is not sufficiently described.

I would give a larger pictures of DECT clinical approved uses, especially in MSK imaging, including papers dealing with the diagnosis of BME, and the evaluation of prosthesis. 

The first topic, also, is important because there are some papers using DECT numbers with relative cut-off for differential diagnosis. These papers could be briefly described in the introduction, included in reference list, and eventually discussed in the discussion section.

Response: We thank the reviewer for this suggestion. We have added a section on clinical applications of DECT imaging in the introduction section (e.g., diagnosis of gout, and bone marrow edema) (lines 38-42). In addition, we have increased the citations of studies examining the utility of DECT numbers in differentiating malignant from benign lesions and included literature-based values for each of these studies (lines 43-86).

Comment 7

After improving the background, I suggest to rephrase the second part of intro section, focusing on rationale of paper.

For example

Some literature-based values of DECT numbers could be used in routine clinical practice for distinguish tumor and metastases from normal tissues and benign lesions.

Response: We thank the reviewer for this suggestion. We have added the following sentence in the introduction section (lines 85-86): “These literature-based values of DECT numbers for distinguishing metastases from non-metastatic lesions could be used in routine clinical practice.”

Comment 8

Rationale is relatively strong to me and should be clarified. 

Response:

We thank the reviewer for this comment. We have re-written the introduction more carefully based on your comment (lines 95-108).

Comment 9

Purpose should be rephrased to be more direct and concise and should be reported at the end of abstract.

Response: We thank the reviewer for this comment. We have rephrased the purpose concisely as follow (lines 108-109): “to investigate the differences in DECT numbers within the upper abdominal organs.” In addition, we have perfectly matched the purpose of the abstract and the introduction sections.

METHODS

Ok

Comment 10

RESULTS

Are overall well presented

To help the reader give main results in the text and leave details in tables.

Response: We thank the reviewer for this advice. We have kept the main results in the results section and removed the remaining ones accordingly.

Comment 11

DISCUSSION

The authors described again results In details here. However my suggestion is to give main results and try a comparison and discussion with previous similar studies. 

Response: We thank the reviewer for this suggestion. To our knowledge, few studies have investigated the similarity between DECT numbers of two entities, such as primary and suspected metastatic lesions. Therefore, it is difficult to compare this study with previous studies. However, in the first paragraph of the discussion section (lines 354-371), we have described several studies that used “simple” DECT numbers to assess lymph node metastasis. In addition, we have featured a study in the third paragraph of the discussion section, which was similar to our study and compared it accordingly (lines 386-399).

Comment 12

Also clinical implications od f results should be discussed here.

Response: We thank the reviewer for your advice. We have carefully re-written the discussion of the clinical implications of our results (lines 363-371, 383-385, 386-399).

Comment 13

Among limitations I would include the use of a single specific scanner (DECT have several type of scanners from different vendors)

Response: We thank the reviewer for this comment. We have accordingly added this content to the limitations section (lines 402-404)

Comment 14

Conclusions are too long; please focus on my results

Response: We thank the reviewer for this suggestion. We have changed the conclusion to be more concise as follows (lines 412-414): “Using the variability in DECT numbers in the same organ as a criterion for defining similarity may be helpful in making a differential diagnosis by comparing DECT numbers of two entities.”

Comment 15

References 

The list is relatively short; authors could search the literature and identify DECT study to give a larger pictures. 

The papers to be considered, not in specific order, are the following.

D'Angelo T, Albrecht MH, Caudo D, Mazziotti S, Vogl TJ, Wichmann JL, Martin S, Yel I, Ascenti G, Koch V, Cicero G, Blandino A, Booz C. Virtual non-calcium dual-energy CT: clinical applications. Eur Radiol Exp. 2021 Sep 3;5(1):38. doi: 10.1186/s41747-021-00228-y. PMID: 34476640; PMCID: PMC8413416.   Yang Z, Zhang X, Fang M, Li G, Duan X, Mao J, Shen J. Preoperative Diagnosis of Regional Lymph Node Metastasis of Colorectal Cancer With Quantitative Parameters From Dual-Energy CT. AJR Am J Roentgenol. 2019 Jul;213(1):W17-W25. doi: 10.2214/AJR.18.20843. Epub 2019 Apr 17. PMID: 30995087.   Diagnostic accuracy of dual-energy CT and virtual non-calcium techniques to evaluate bone marrow edema in vertebral compression fractures G Foti, A Beltramello, M Catania, S Rigotti, G Serra, G Carbognin La radiologia medica 124 (6), 487-494     Identification of bone marrow edema of the ankle: diagnostic accuracy of dual-energy CT in comparison with MRI G Foti, M Catania, S Caia, L Romano, A Beltramello, C Zorzi, G Carbognin La radiologia medica 124 (10), 1028-1036   · Preoperative Prediction of Cervical Nodal Metastasis in Papillary Thyroid Carcinoma: Value of Quantitative Dual-Energy CT Parameters and Qualitative Morphologic Features. Wu YY, Wei C, Wang CB, Li NY, Zhang P, Dong JN.AJR Am J Roentgenol. 2021 May;216(5):1335-1343. doi: 10.2214/AJR.20.23516. Epub 2021 Mar 24.PMID: 33760651   · Clinical significance of dual-energy CT-derived iodine quantification in the diagnosis of metastatic LN in colorectal cancer.Kato T, Uehara K, Ishigaki S, Nihashi T, Arimoto A, Nakamura H, Kamiya T, Oshiro T, Ebata T, Nagino M.Eur J Surg Oncol. 2015 Nov;41(11):1464-70. doi: 10.1016/j.ejso.2015.08.154. Epub 2015 Aug 22.PMID: 26329783

Response: We thank the reviewer for this advice. References have been added, including the above. The following new references were included: [9], [10], [11], [12], [13], [14], [15], [16], [19], and [21].

Tables ok

Figures ok

Response: We thank the reviewer for this kind comment.

Reviewer 2 Report

Abstract:

The authors reveal the main problem and highlight the main finding of the study in the abstract. It includes all the necessary information.

Introduction:

This section details the state-of-the-art of the dual-energy CT. The quality of the cited papers is good. However, it is highly recommended to extend the number of cited studies. This would increase the scientific soundness of the paper.

The goal of the study and the hypothesis of the authors are clear. The contribution of the study is added.

Materials and methods:

The study includes no materials, the title should be shortened to methods.

The description of the population is comprehensive, the information support the replicability of the study. The protocol, the image analysis, and the statistical analysis methods are also clear and support the replicability.

The methodological part of the study is profoundly elaborated.

Results / Discussion:

The obtained information are well-presented. There is no conceptual error in the provided results, and they are explained well. The tables and figures are simple and clear, they support understanding the findings.

I recommend redrafting Fig 3 (Increasing the main information (6 diagrams) by putting the legend below them. I recommend decreasing the internal captions (a, b, c, d) of Fig 4.

The discussion section is good.

Conclusion:

I recommend summarizing the main finding shortly in the form of ’take-away’ sentences (bullet points) that can be understood without reading the whole paper.

Language:

The language is outstanding, it’s style is formal academic writing. No errors were detected.

Author Response

REVIEWER COMMENTS:

Reviewer #2:

This is the review of the paper 

Can dual-energy computed tomography data-based quantitative parameters be used for differential image diagnosis? A 3 preliminary study on the variability of dual-energy computed  tomography parameters in healthy upper abdominal organs

Manuscript ID: tomography-1957996

We hereby resubmit our manuscript titled, “DECT numbers in upper abdominal organs for differential diagnosis: a feasibility study.”

We sincerely appreciate all the comments and suggestions from the three reviewers as they have helped make our manuscript more precise and informative. We have read all the comments carefully and accordingly revised our manuscript.

Our point-by-point responses to each comment are noted below.

We really appreciate your kind consideration on this manuscript for publication in Tomography.

Comment 1

Introduction:

This section details the state-of-the-art of the dual-energy CT. The quality of the cited papers is good. However, it is highly recommended to extend the number of cited studies. This would increase the scientific soundness of the paper.

The goal of the study and the hypothesis of the authors are clear. The contribution of the study is added.

Response: We thank the reviewer for this advice. We have increased the citations of studies examining the utility of DECT accordingly (lines 38-84). The following new references were included: [9], [10], [11], [12], [13], [14], [15], [16], [19], and [21].

Comment 2

Materials and methods:

The study includes no materials, the title should be shortened to methods.

The description of the population is comprehensive, the information support the replicability of the study. The protocol, the image analysis, and the statistical analysis methods are also clear and support the replicability.

The methodological part of the study is profoundly elaborated.

Response: We thank the reviewer for this suggestion. We have shortened the section title to “Methods” accordingly.

Comment 3

Results / Discussion:

The obtained information are well-presented. There is no conceptual error in the provided results, and they are explained well. The tables and figures are simple and clear, they support understanding the findings.

I recommend redrafting Fig 3 (Increasing the main information (6 diagrams) by putting the legend below them. I recommend decreasing the internal captions (a, b, c, d) of Fig 4.

The discussion section is good.

Response: We thank the reviewer for this suggestion. We have modified Fig 3 (page 7) and Fig 4 (page 9) accordingly.

Comment 4

Conclusion:

I recommend summarizing the main finding shortly in the form of ’take-away’ sentences (bullet points) that can be understood without reading the whole paper.

Response: We thank the reviewer for this recommendation. We have added “take away in a sentence” after the conclusion accordingly (lines 415-528)

Language:

The language is outstanding, it’s style is formal academic writing. No errors were detected.

Response: We thank the reviewer for this kind comment.

Reviewer 3 Report

The paper describes a preliminary study act to evaluate the capability to distinguish between different tissues (healthy or not)  uning 2 energies CT. It is well recognized  the strong effort of archival research and the importance of finding a method based on this tecnique.  The paper needs to be better presented, in this fashion it looks difficoult to be read. Whole sections are a repetition of numbers already presented in table that does not help the reader to understanding the central idea of the paper. Finally I would like to rise some question or consideration: 

As far I understand Fig 2 is taken from one single out of 30 patients. All other patients ROI were taken similarly?  WHat is the criteria of selecting ROIs in different organs of different patients? Are the ROI taken in similar areas? I also see in liver white regions that have been excluded. Can the author describe them? 

Caption of Fig 2 is excessively long and not clear what it the meaning of  it. What it is supposed to be dimonstrated with the comparison between the ROIs of two different regions of the same tissue? Furthermore I do not unserstand what is the sum or average of a ROI (line 119-120). The author means the average of the counts within the ROI? ARe these images at  40 keV or 70 keV? Or is it the difference?

What is the mean age of the patients useful for? Furthermore if the real range is so large 22-80 years is there any significant variation in the method validity for young or older tissue?

Paragraph 3.2: there is no need to report in the text the same values that are reported in a table. The text is badly readable and unuseful. 

is very difficoult in Fig 4 to understand the  line corresponding to the same organs and these corresponding to the different organs.

Author Response

REVIEWER COMMENTS:

Reviewer #3:

This is the review of the paper 

Can dual-energy computed tomography data-based quantitative parameters be used for differential image diagnosis? A 3 preliminary study on the variability of dual-energy computed  tomography parameters in healthy upper abdominal organs

Manuscript ID: tomography-1957996

We hereby resubmit our manuscript titled, “DECT numbers in upper abdominal organs for differential diagnosis: a feasibility study.”

We sincerely appreciate all the comments and suggestions from the three reviewers as they have helped make our manuscript more precise and informative. We have read all the comments carefully and accordingly revised our manuscript.

Our point-by-point responses to each comment are noted below.

We really appreciate your kind consideration on this manuscript for publication in Tomography.

Comment 1

The paper describes a preliminary study act to evaluate the capability to distinguish between different tissues (healthy or not)  uning 2 energies CT. It is well recognized  the strong effort of archival research and the importance of finding a method based on this tecnique.  The paper needs to be better presented, in this fashion it looks difficoult to be read. Whole sections are a repetition of numbers already presented in table that does not help the reader to understanding the central idea of the paper. Finally I would like to rise some question or consideration: 

Response: We thank the reviewer for this comment. The complete manuscript has been re-written to improve readability. In addition, we have removed most of the results presented in the tables.

Comment 2

As far I understand Fig 2 is taken from one single out of 30 patients. All other patients ROI were taken similarly?  WHat is the criteria of selecting ROIs in different organs of different patients? Are the ROI taken in similar areas? I also see in liver white regions that have been excluded. Can the author describe them? 

Response: We thank the reviewer for these comments. We have re-written the description of ROI placement more carefully in the “2.3. Collection of DECT numbers” section (lines 155-177). It is correct that Fig 2 is taken from one of the 30 patients. ROIs of the remaining 29 patients were similarly placed. Six ROIs were marked on each non-contrast enhanced and contrast-enhanced triple-phasic CT images for 30 patients, avoiding vessels (white regions?), bile ducts, and the renal calyx. The locations of the six ROIs were as follows: ROI 1, the left hepatic lobe; ROI 2, right hepatic lobe; ROI 3, the dorsal side of the spleen; ROI 4, the ventral side of the spleen; ROI 5, right renal parenchyma; and ROI 6, left renal parenchyma. A GSI viewer used in this study automatically calculated various DECT numbers for the ROI when the ROI was placed on a 70 keV virtual monochromatic image. In this study, DECT numbers of six quantitative parameters, including (a) 70 keV CT value, (b) 40 keV CT value, (c) slope, (d) effective Z, (e) iodine concentration, and (f) water concentration, were collected for each non-contrast enhanced and contrast-enhanced triple-phasic CT image. Therefore, A total of 4,320 DECT numbers (30 patients × 4 phasic CT images × 6 locations × 6 DECT numbers) were finally collected.

Comment 3

Caption of Fig 2 is excessively long and not clear what it the meaning of  it. What it is supposed to be dimonstrated with the comparison between the ROIs of two different regions of the same tissue? Furthermore I do not unserstand what is the sum or average of a ROI (line 119-120). The author means the average of the counts within the ROI? ARe these images at  40 keV or 70 keV? Or is it the difference?

Response: We thank the reviewer for this comment. We have modified the caption of Fig 3 accordingly (lines 179-188). In addition, the methods of analysis of DECT numbers obtained from ROIs have been removed from the caption of Fig 2. Instead, we have added a new paragraph “2.4. Analyses of DECT numbers” in the method section (lines 189-202). In this study, we examined two points: i) the agreement (variability) of DECT numbers and ii) the diagnostic ability using the differences in DECT numbers. Regarding i), we assessed agreement in DECT numbers within the same health upper abdominal organs (liver, spleen, or kidney). In other words, we evaluated the agreement between DECT numbers derived from ROI 1 and 2 (liver), ROI 3 and 4 (spleen), and ROI 5 and 6 (kidney), respectively. Regarding ii), we assessed the diagnostic ability of each DECT parameter to differentiate measurements taken from the same organs and measurements taken from the different organs. Therefore, ROC analysis was performed considering the differences in DECT numbers obtained from ROI 1 and 2 (liver), ROI 3 and 4 (spleen), and ROI 5 and 6 (kidney) as true (same organ); and the differences in DECT numbers obtained from ROI 1 and 3 (liver and spleen), ROI 3 and 5 (spleen and kidney), and ROI 5 and 1 (kidney and liver) as false (different organ). Finally, Fig 2 was a 70-keV CT image. A GSI viewer used in this study automatically calculated various DECT numbers including the 40-keV CT value for the ROI when the ROI was placed on a 70-keV image.

Comment 4

What is the mean age of the patients useful for? Furthermore if the real range is so large 22-80 years is there any significant variation in the method validity for young or older tissue?

Response: We thank the reviewer for this comment. Because of the retrospective design of our study (various CT indications), patients of various age groups (young to elderly) were included in this study. As shown below, age was not associated with any DECT numbers (line 246).

  • 70-keV CT value: r2 < .001; p = .88
  • 40-keV CT value: r2 < .001; p = .89
  • slope: r2 < .001; p = .90
  • effective Z: r2 < .001; p = .96
  • iodine concentration: r2 < .001; p = .90
  • water concentration: r2 < .001; p = .97

Comment 5

Paragraph 3.2: there is no need to report in the text the same values that are reported in a table. The text is badly readable and unuseful. 

Response: We thank the reviewer for this suggestion. We have removed most of the results presented in Table 1 accordingly (line 253-258).

Comment 6

is very difficoult in Fig 4 to understand the  line corresponding to the same organs and these corresponding to the different organs.

Response: We thank the reviewer for this comment. To make Fig 4 easier to understand, we have described “2.4. Analyses of DECT numbers” (lines189-202) more carefully in the Methods section and modified “3.3 Diagnostic ability” in the Results section (lines 292-313).

Response: We thank the reviewer for this kind comment.

Round 2

Reviewer 3 Report

The authors have responded to all the questions and suggestions that I have posed. The paper has been improved considerably. I recommend the paper for publication